# Modeling the Formation of Gas Bubbles inside the Pores of Reactive Electrochemical Membranes in the Process of the Anodic Oxidation of Organic Compounds

**DOI:** 10.3390/ijms22115477

**Published:** 2021-05-22

**Authors:** Semyon Mareev, Ekaterina Skolotneva, Marc Cretin, Victor Nikonenko

**Affiliations:** 1Physical Chemistry Department, Kuban State University, 149 Stavropolskaya st., 350040 Krasnodar, Russia; mareev-semyon@bk.ru (S.M.); ek.skolotneva@gmail.com (E.S.); 2Institut Européen des Membranes-UMR5635, 34095 Montpellier, France; marc.cretin@umontpellier.fr

**Keywords:** reactive electrochemical membrane, porous electrode, gas bubbles, anodic oxidation, hydroxyl radicals

## Abstract

The use of reactive electrochemical membranes (REM) in flow-through mode during the anodic oxidation of organic compounds makes it possible to overcome the limitations of plate anodes: in the case of REM, the area of the electrochemically active surface is several orders of magnitude larger, and the delivery of organic compounds to the reaction zone is controlled by convective flow rather than diffusion. The main problem with REM is the formation of fouling and gas bubbles in the pores, which leads to a decrease in the efficiency of the process because the hydraulic resistance increases and the electrochemically active surface is shielded. This work aims to study the processes underlying the reduction in the efficiency of anodic oxidation, and in particular the formation of gas bubbles and the recharge of the REM pore surface at a current density exceeding the limiting kinetic value. We propose a simple one-dimensional non-stationary model of the transport of diluted species during the anodic oxidation of paracetamol using REM to describe the above effects. The processing of the experimental data was carried out. It was found that the absolute value of the zeta potential of the pore surface decreases with time, which leads to a decrease in the permeate flux due to a reduction in the electroosmotic flow. It was shown that in the solution that does not contain organic components, gas bubbles form faster and occupy a larger pore fraction than in the case of the presence of paracetamol; with an increase in the paracetamol concentration, the gas fraction decreases. This behavior is due to a decrease in the generation of oxygen during the recombination reaction of the hydroxyl radicals, which are consumed in the oxidation reaction of the organic compounds. Because the presence of bubbles increases the hydraulic resistance, the residence time of paracetamol—and consequently its degradation degree—increases, but the productivity goes down. The model has predictive power and, after simple calibration, can be used to predict the performance of REM anodic oxidation systems.

## 1. Introduction

Over the past decades, the amount of organic waste, including biorefractory waste, has grown exponentially. Up to ten percent of these pollutants end up in wastewater, which makes the problem of removing organic compounds—which are resistant to traditional biological methods of wastewater treatment—urgent. Pharmaceuticals [1,2] and petroleum products, including microplastics [3,4], cannot be captured by existing municipal wastewater treatment plants. Advanced electrochemical oxidation processes, and anodic oxidation in particular [5,6], are the most promising solutions, and are considered by the scientific community to be next-generation technologies for the purification of waste and natural waters from organic pollutants.

The anodic oxidation process allows one to mineralize organic compounds through a combination of direct electron transfer and oxidation by hydroxyl radicals, which are generated during the water discharge on the anode surface and are highly reactive to most toxic organic pollutants [7]. Most of the organic substances contained in wastewater, including aromatic compounds, undergo mineralization [8,9,10], and with certain parameters of the system, one can achieve 100% degradation, but at the same time, energy consumption naturally increases [11,12].

However, there are a number of limitations that prevent the widespread utilization of anodic oxidation in practice. The main problem is diffusion restrictions: hydroxyl radicals are formed only on the anode surface and have a short lifetime, and therefore are present only in a thin boundary layer (<1 μm) [6]. As a result, the oxidation of the organic pollutants occurs at the electrode surface, and the process is limited by the convective-diffusion delivery of pollutants from the bulk of the solution to the reaction zone. There are several known ways to solve this problem, and the most effective one is to change the hydrodynamics of the electrochemical cell in order to increase the mass transfer characteristics. In [13], it was shown that the utilization of porous electrodes, also called “reactive electrochemical membranes” (REM), in cross-flow configurations allows us to overcome the diffusion restrictions.

The gas bubble evolution on the electrode surface is a phenomenon that accompanies most electrochemical processes, in which the direct electron transfer reaction leads to the formation of products with limited solubility in the electrolyte. In the process of the anodic oxidation on REM, gas bubbles are generated both on the surface of the anode and in its bulk. They are formed as a result of water discharge when the electrode potential exceeds the oxygen evolution potential (OEP), or as a result of other electrochemical or chemical reactions (for example, the generation of CO_2_ during the oxidation of organic compounds). This phenomenon negatively affects the efficiency of the process, because gas bubbles shield the electroactive surface of the electrode and prevent the molecules of the target pollutant from entering the reaction zone. They also reduce the electrical conductivity of the solution, which leads to an increase in energy consumption and the hydrodynamic resistance of the system as a result of the partial or complete blockage of the pore space. However, a number of studies have shown that gas bubbles can play a positive role in the anodic oxidation process. For example, they can act as a physical barrier to prevent fouling [14] or increase the rinsing efficiency by raising the contaminant layer [15,16].

Mathematical modeling is the most important way to deepen the understanding of the behavior of physical systems. Despite the extensive modeling of bubble evolution from flat surfaces [17,18], there is relatively little theoretical work on the formation of bubbles inside porous electrodes [19]. There is even less work on the modeling of the evolution of gas bubbles in systems with a cross-flow configuration when a solution is pumped through the anode. In the case of the anodic oxidation of organic pollutants, we have not found theoretical works devoted to this problem that take into account the reactions of organic compound oxidation.

We present a simple one-dimensional non-stationary mathematical model of the oxygen bubble evolution inside a porous electrode during the anodic oxidation of paracetamol. The purpose of this work is to assess the contribution of gas bubbles in the pores of REM to the productivity of the anodic oxidation process, as well as to identify the other processes responsible for a decrease or increase in the resulting permeate flux.

## 2. Mathematical Model

The system under study consists of three layers: a diffusion layer with a thickness of *δ*, a porous electrode with a thickness of *d*, and a permeate solution (Figure 1). We consider the transfer of four particles: hydroxyl radicals (HO^•^), paracetamol (PCT) molecules, the by-products of PCT oxidation (BP), and molecular oxygen (O_2_). The electrolysis system is considered in a crossflow filtration mode operated under galvanostatic conditions. The PCT solution with an initial concentration of *c*_0_ enters the system under study in the *x* = −*δ* coordinate, flows in the direction of the coordinate axis, and moves away (the permeate flux) in the *d* coordinate.

According to previous theoretical investigations [20,21,22], the following assumptions can be made:The transport numbers of HO^•^, molecular O_2_, PCT and BP are negligible compared to the transport number of the supporting electrolyte, and only their diffusion and convection fluxes are taken into account.The charging current is not considered; only the Faraday current is taken into account.The oxidation reaction of organic compounds by the direct electron transfer on the REM surface is neglected.Temperature and activity coefficients are ignored.

Hydroxyl radicals are generated on the surface of the anode from water discharge, according to the following reaction [21]:(1)$$H_{2}O⇄{\mathrm{HO}}_{\left(aq\right)}^{\text{•}}+H_{\left(aq\right)}^{+}+e^{-}$$

Free hydroxyl radicals can react with each other to form hydrogen peroxide:(2)$${\mathrm{HO}}^{\text{•}}+{\mathrm{HO}}^{\text{•}}\overset{k_{{\mathrm{HO}}^{\text{•}}}}{\to }H_{2}O_{2}$$
Here, *k*_HO_^•^ is the rate constant of the reaction (2).

Hydrogen peroxide can react with hydroxyl radicals to form oxygen:(3)$$H_{2}O_{2}{+2\mathrm{HO}}^{\text{•}}\overset{k_{H_{2}O_{2}}}{\to }{2H}_{2}{O+O}_{2}$$
Here, $k_{H_{2}O_{2}}$ is the rate constant of the reaction (3).

The rate constant of the reaction (3) is several orders of magnitude higher than that of reaction (2). Because reaction (2) is limiting, (2) and (3) can be rewritten as:(4)$${4\mathrm{HO}}^{\text{•}}\overset{k_{{\mathrm{HO}}^{\text{•}}}}{\to }{2H}_{2}{O+\text{O}}_{2}$$

In [23], the authors state that hydroxyl radicals and hydrogen peroxide are the most likely products of the water discharge on the surface of an anode made of titanium oxide. However, at high overpotentials (as in our case), the oxygen evolution reaction proceeds preferably. In the current work, we consider the following oxygen evolution reaction:(5)$${2H}_{2}O\to O_{2}{+4H}^{+}+4e$$

The hydroxyl radicals react with the organic compound (in our case it is paracetamol, PCT) and its degradation by-products, in parallel to reaction (4):(6)$$\mathrm{PCT}\overset{k_{\mathrm{PCT}}}{\underset{\mathrm{HO}\text{•}}{\to }}\mathrm{by}-\mathrm{products}\overset{k_{i}}{\underset{\mathrm{HO}\text{•}}{\to }}\mathrm{end}-\mathrm{products}$$
Here, $k_{\mathrm{PCT}}$ is the PCT oxidation rate constant and *k*_i_ is the oxidation rate constant of *i*th byproduct.

According to [20,24], the rate expression for a second-order reaction (2) is as follows:(7)$$r_{{\mathrm{HO}}^{\text{•}}}=k_{{\mathrm{HO}}^{\cdot }}c_{{\mathrm{HO}}^{\cdot }}^{2}$$
Here, $c_{{\mathrm{HO}}^{\cdot }}^{}$ is the concentration of the hydroxyl radicals.

It is assumed that the organic compound molecule decomposes as soon as it reacts with one hydroxyl radical. Then, the reaction rate of its oxidation will be:(8)$$r_{\mathrm{PCT}}=k_{\mathrm{PCT}}c_{\mathrm{PCT}}c_{{\mathrm{HO}}^{\text{•}}}$$
Here, *c*_PCT_ is the concentration of PCT.

Each byproduct in reaction (6) has its own oxidation rate constant, *k*_i_, corresponding to its reactivity with hydroxyl radicals [24]. The consideration of the reaction of each byproduct significantly complicates the description of the kinetics of the process. In our previous work [25] we proposed an approach that makes it possible to simplify the solution of the mathematical problem: in the calculations, we used the average $k_{\mathrm{BP}}$ and integral *c*_BP_ values. Thus, the rate of degradation of the by-products reads:(9)$$r_{\mathrm{BP}}=\sum_{i=1}^{n}k_{i}c_{{\mathrm{HO}}^{\text{•}}}c_{i}=k_{\mathrm{BP}}c_{{\mathrm{HO}}^{\text{•}}}c_{\mathrm{BP}}$$

The current density spent on the direct electron transfer reactions (1) and (5) can be expressed using the Butler–Volmer equation:(10)$$i_{k}^{}=i_{0}^{k}\exp\left[\frac{-\beta n_{k}F}{RT}\eta_{}^{k}\right]$$
(11)$$\eta_{}^{k}=\varphi_{m}-\varphi_{s}-E_{k}^{0}$$
Here, $i_{k}^{}$, $i_{0}^{k}$, $n_{k}$, $\eta_{}^{k}$, $E_{k}^{0}$ are the total current, exchange current, number of electrons, overpotentials and formal potentials in reactions (1) and (5) of the *k*th species, where *k* takes the values of HO^•^ and O_2_; β is the symmetry coefficient; *F* is the Faraday constant, *R* is the gas constant, *T* is the temperature, and $\varphi_{s}$ and $\varphi_{m}$ are potentials in the phase of the solution (index *s*) and the material (index *m*) of the anode, respectively.

The transport of diluted species in the solution is described by the equation system, which consists of Fick’s law with the convective term (12), the material balance Equation (13), Ohm’s law in the differential form written for each of the phases (14)–(15), the charge conservation law (16)–(17), Darcy’s law (18) with the electroosmotic term (19), and the function of the dependence of the gas volume fraction on the concentration of O_2_ (20):(12)$${\vec{J}}_{k}=-\varepsilon_{s}D_{k}\frac{\partial c_{k}}{\partial x}+c_{k}\vec{v}$$
(13)$$\varepsilon_{s}\frac{\partial c_{k}}{\partial t}=-\mathrm{div}{\vec{J}}_{k}+\varepsilon_{s}R_{k}$$
(14)$$i_{s}=-\varepsilon_{s}\kappa_{s}\frac{\partial \varphi_{s}}{\partial x}$$
(15)$$i_{m}=-\varepsilon_{m}\kappa_{m}\frac{\partial \varphi_{m}}{\partial x}$$
(16)$$\frac{\partial (i_{s}+i_{m})}{\partial x}=0$$
(17)$$\frac{\partial (i_{s})}{\partial x}=-\varepsilon_{s}\sum_{k=1}^{2}a_{v}i_{k}$$
(18)$$\vec{v}={\left(\frac{\varepsilon_{s}}{\varepsilon_{p}}\right)}^{\frac{2}{3}}\left(-\frac{\sigma }{\mu }\frac{\partial p}{\partial x}+{\vec{v}}_{EO}\right)$$
(19)$${\vec{v}}_{EO}=\frac{\varepsilon \varepsilon_{0}\zeta }{\mu }\frac{\partial \varphi_{s}}{\partial x}$$
(20)$$\frac{\partial \varepsilon_{g}}{\partial t}=k_{v}\left(\varepsilon_{s}c_{O_{2}}^{s}k_{g}-\varepsilon_{g}\right)$$
Here, ${\vec{J}}_{k}$ and $D_{k}$ are the flux density and diffusion coefficient of the *k*th component of the system, respectively; $i_{j}$, $\kappa_{j}$ and $\varepsilon_{j}$ are the current density, conductivity and volume fraction of the solution (*j = s*) and the anode material (*j = m*); *a*_v_ is the specific surface area of the electrode;$\vec{v}$ is the total velocity of the permeate; ${\vec{v}}_{EO}$ is the velocity of the electroosmotic flow; *σ* is the permeability coefficient; *p* is the pressure; *k*_v_ is the mass coefficient characterizing the rate of the transition of oxygen from the gas phase (index *g*) into the solution (index *s*) and vice versa; *k*_g_ is the volatility coefficient; $\varepsilon_{g}=\varepsilon_{p}-\varepsilon_{s}$ is the gas fraction and $\varepsilon_{p}$ is the pore fraction; *ε_0_* is the vacuum permittivity; *ε* is the relative permittivity; *ζ* is the zeta potential of the pore surface; *μ* is the dynamic viscosity; *R_k_* is the sum of the reaction rates leading to the appearance or removal of the *k*th component of the system.

For hydroxyl radicals, we assume that four of them are spent on the formation of one molecule of oxygen, one on the oxidation of PCT and *z*_BP_ on the oxidation of the by-products. The concentration of oxygen also changes due to the volatilization process. We use the term $k_{v}\left(\varepsilon_{s}c_{O_{2}}^{s}-\frac{\varepsilon_{g}}{k_{g}}\right)$ to describe the process. Thus, the total reaction rates read:(21)$$R_{{\mathrm{HO}}^{•}}=-4r_{{\mathrm{HO}}^{•}}-r_{\mathrm{PCT}}-z_{\mathrm{BP}}r_{\mathrm{BP}}-\frac{i_{{\mathrm{HO}}^{•}}a_{v}}{n_{{\mathrm{OH}}^{•}}F}$$
(22)$$R_{\mathrm{PCT}}=-r_{\mathrm{PCT}}$$
(23)$$R_{\mathrm{BP}}=r_{\mathrm{PCT}}-r_{\mathrm{BP}}$$
(24)$$R_{O_{2}}=r_{{\mathrm{HO}}^{•}}-k_{v}\left(\varepsilon_{s}c_{O_{2}}^{s}-\frac{\varepsilon_{g}}{k_{g}}\right)-\frac{i_{O_{2}}a_{v}}{n_{O_{2}}F}$$

The sum of the current densities (*i*_m_ + *i*_s_) in the system is assumed to be constant and equal to *i*_tot_.

In the bulk of the solution, (x=−δ), the concentration of paracetamol, the electrolyte potential and the pressure are a set constant:(25)$$c_{\mathrm{PCT}}=c_{0}$$
(26)$$\varphi_{s}=0$$
(27)p=0

In the outlet of the REM (*x = d*) the permeate flux is equal to the convective term and the current density in the electrode material phase is equal to the total. Thus, we can write the following boundary conditions:(28)$$\frac{dc_{PCT}}{dx}=0$$
(29)$$\frac{dc_{O_{2}}}{dx}=0$$
(30)$$i_{m}=i_{tot}$$
(31)p=TMP
Here, TMP is the applied transmembrane pressure.

We used the following initial conditions (*t* = 0) for the concentrations in the system:(32)$$c_{PCT}=c_{0}$$
(33)$$c_{{\mathrm{HO}}^{•}}=0$$
(34)$$c_{O_{2}}=0$$

## 3. Results and Discussions

The mathematical problem was solved numerically using the Comsol Multiphysics 5.5 software package.

### 3.1. Degradation of Paracetamol and Its Byproducts

We consider the mineralization of paracetamol as a target chemical reaction in the calculations. In our previous work [25], we took into account the decrease in the concentration of HO^•^ radicals due to the oxidation reaction of the byproducts. However, the concentration of the byproducts was considered indirectly. Brillas and coauthors [26,27] proposed possible methods of PCT oxidation by HO^•^. In the present work, it was suggested to take hydroquinone as the main byproduct, which is strongly refractory to direct electron transfer. The end-products, such as carboxylic acids, have a lower oxidation rate by HO^•^, and faster mineralization can be achieved by DET [28], but in current work, these reactions are not taken into account. Therefore, it was calculated that 27 hydroxyl radicals can participate in the oxidation reaction of hydroquinone into carboxylic acids, and we used the number in all of the calculations presented in this study.

In numerous articles [29,30,31,32], the value of the rate constant of the PCT oxidation reaction *k*_PCT_ differs in the range between 2 × 10^6^ and 1.4 × 10^7^ mol·s^−1^m^3^. We used an average value of *k*_PCT_ = 1 × 10^7^ mol·s^−1^m^3^ (Table 1). The value of *k_BP_* and *D*_BP_ were considered to be equal to those of hydroquinone, as a main byproduct of the PCT degradation reaction.

### 3.2. The Experimental Data

The experiments were carried out using a cross-flow electrolyzer, which utilizes REM as a porous anode in inside–outside cross-flow filtration mode. The experimental data were taken from [30]. The REM is a porous electrode (in our case, it is a tubular Magnelli phase sub-stoichiometric titanium oxide electrode). The transmembrane pressure was used as an independent parameter of the experiment. The feed solution contains a PCT of varying concentrations, which is the target pollutant, and a supporting electrolyte (0.1 M Na_2_SO_4_), which presumably does not participate in any chemical reactions and is used only to decrease the total resistance of the system. The following current mode was used during the anodic oxidation processes: the current density was set at −300 A/m^2^ for 90 min, then, no current was supplied for 50 min. During the experiment, the permeate flux was measured. The parameters of the experimental setup, which are essential for the calculations, were as follows: the REM thickness was *d* = 2 mm, the transmembrane pressure was constant and equal to 40 mbar, the TOC_PCT_ was 17 or 141 mgC/L, the concentration of the supporting electrolyte (sodium sulfate) was 50 mM. The REM has a monomodal pore size with an average value 1.4 μm, and a relative porous volume of *ε* = 0.41. The value of *δ =* 30 μm was obtained in [25] using the Lévêque approximate solution for the hydrodynamic conditions of the electrolyzer used in the experiments.

### 3.3. Electroosmotic Flow

The experimental data shows (Figure 2) that the permeate flux through the REM sharply increases immediately after the electric current is switched on. This behavior is typical for porous structures with a nonzero surface charge: in the presence of a potential difference at the input and output of the electrode, electroosmotic flow arises along the pore walls, which sets in motion the entire intrapore solution. The surface charge is characterized by the *ζ*-potential presented in the Helmholtz–Smoluchowski equation (Equation (19)). In our case, according to the available studies of the surface charge of the pores of ceramic membranes [37], the value of the *ζ*-potential should be in the range from 0 to −40 mV. If the potential difference at the electrode boundaries is equal to 0.5 V (the value calculated using the model) then the value of *ζ* should be about −20 mV in order to obtain a permeate flux corresponding to the experimental data.

As can be seen in Figure 2, at the initial time, there is a good agreement between the experimental and calculated data at all of the concentrations of paracetamol. At zero or low PCT concentrations (17.7 mgC/L), the calculated and experimental data agree up to *t* = 30 min. After that, the experimental permeate flux gradually decreases, which may indicate a change in the charge of the pore surface and a decrease in the electroosmotic flow. In our model, we do not take into account the decrease in the absolute value of the *ζ* potential; therefore, after the electric current is turned off (*t* = 90 min), the calculated electroosmotic flow sharply decreases and the permeate flux agrees with the experimental data. In the presence of PCT, additional processes occur on the surface of the REM pores (e.g., the adsorption of organic molecules), which lead to a decrease in the absolute value of zeta potential and, therefore, a more significant deceleration of the electroosmotic flow compared to the case without PCT. When the concentration of paracetamol is 141 mgC/L, the experimental values of the permeate flux are slightly scattered in time, which is probably due to unsteady processes of anodic oxidation and oscillations of the zeta potential value, but the general trend remains unchanged. Furthermore, as in the cases of PCT = 0 and PCT = 17.7 mgC/L, the absolute value of the zeta potential decreases with time, and at the moment when the electric current is turned off (*t* = 90 min), the theoretical and experimental values of the permeate flux are in good agreement.

### 3.4. The Bubble Formation

There are several cases in which gas bubbles may occur in the pores of REM. The first and foremost is the oxygen evolution reaction. In this case, the overpotential on the electrode must be sufficient for this reaction to occur, and the electrode must be catalytically active for the generation of oxygen at a given current density. In the case of the Magnelli phase anode used in the current work, this catalytic activity appears at relatively high overpotentials. Therefore, we consider the current, which is several times higher than the limiting value. The mechanisms of the bubble formation are comprehensively described in [19]. The TiOx Magnelli phase anode is a hydrophilic material, and therefore the bubbles relatively weakly touch the pore surface (Figure 3). In this state, the bubble can grow continuously, and if it is not removed along with the flow of the solution, it is able to close up the pore quickly enough. As a result, some of the pores can be completely blocked and the flow is redistributed through the free pores, but the resulting permeate flux is reduced. On the other hand, due to the hydrophilicity of the anode material, some of the bubbles can freely move along with the core of the solution flow. The rate of change in the bubble size depends on the excess in the concentration of oxygen in the solution, *c*_O2_; the volume fraction of this solution, *ε*_L_; the volatility coefficient, *k*_g_; and the transfer coefficient of the O_2_ molecule across the solution–gas interface, *k*_v_ (Equation (19)). The bubble size grows with an increase in the concentration of O_2_. Over time, the size of the bubble reaches its maximum, partially or completely blocking the pore. The consideration of the shape and size of individual bubbles is a particular problem. In this paper, the average volume fraction of all of the bubbles is considered. The partial or complete blockage of pores was taken into account using the power dependence in Equation (17).

### 3.5. Determination of the Volatility and Mass Transfer Coefficients across the Solution–Gas Interface

These coefficients were considered as fitting parameters (Table 1). The mass transfer coefficient, *k*_v_, characterizes the rate of the transition of oxygen molecules from the dissolved state to bubbles fixed on the surface of the pores. The transition from the dissolved state can also follow a more complex path: first, microbubbles are formed, then they move in the bulk of the pore and collapse with immobile bubbles. It should be noted that this coefficient does not take into account the formation of all of the bubbles freely moving along the volume of the pore space, only those which collapse with immobile ones. As can be seen from Figure 4a, *k*_v_ primarily affects the time at which the system reaches a steady-state, while the value of the steady-state permeate flux does not depend on *k*_v_.

Volatility is a material quality that describes how readily a substance vaporizes. We take this parameter into account through the coefficient *k*_g_, which relates the molar concentration of the dissolved oxygen to its molar concentration in the bubbles. This implicitly includes the molar concentration of oxygen in the bubbles; therefore, the dimension of this coefficient is m^3^/mol. The value of *k*_g_ affects the fraction of space occupied by bubbles, *ε*_g_, and hence the permeate flux (Figure 4b). As can be seen from Figure 4b, with an increase in *k*_g_, the time taken to reach a steady state also increases.

In our calculations, both coefficients were assumed to be constant during the processing of the experimental data, and their values are presented in Table 1. It should be noted that in the presence of PCT, this assumption may be incorrect because the experimental time to reach a steady state after switching off the electric current is much greater than the calculated value (Figure 2). We have shown that the fitting of the parameter allows one to achieve a good agreement between the experimental and calculated data (Figure 4b).

### 3.6. Influence of the Paracetamol Degradation on the Permeate Flux

At high current densities, the degradation of paracetamol proceeds quite intensively and the PCT concentration at the outlet of the REM is equal to zero. At a low PCT concentration (17.7 mgC/L) at a depth of 30 μm of the porous electrode, the 99% degradation of the byproducts is observed (Figure 5a). At a higher concentration (141 mgC/L), the same degradation value is observed after the middle of the REM at a distance of 1200 μm. In our previous work [25] we simulated a stationary degradation process of an aqueous solution with a low concentration of PCT (18 mgC/L) and obtained a similar result: the 99% degradation of the byproducts occurs at a depth of 5 μm. It should be noted that in [25] we did not take into account the oxygen evolution reaction; thus, at the same current density, there was much more HO^•^, which oxidized more molecules of PCT. In the current work, the concentration of the byproducts degraded through the oxidation reaction with HO^•^ also sharply decreases and reaches small values at approximately the same depth of REM as PCT (Figure 5b).

The presence of paracetamol reduces the amount of HO^•^ that can react with itself and form O_2_. As a result, less molecular oxygen is generated and the permeate flux is reduced less (Figure 2). It is worth noting that when PCT is completely mineralized, CO_2_ is formed, which can also volatilize and form gas bubbles. During the mineralization of 141 mgC/L, only 12 mmol/L of CO_2_ could be formed. The solubility limit of carbon dioxide at room temperature and atmospheric pressure is about 30 mmol/L; therefore, in the present work, the CO_2_ remains dissolved in the water and does not participate in the bubble evolution.

As was mentioned above, the formation of byproducts of the PCT oxidation reaction leads to a change in the electroosmotic flow and the scattering of the permeate flux data. At times from 15 to 90 min, the experimental values for the permeate flux are almost always lower than the calculated ones. However, after turning off the current, the flux values exactly coincide at all concentrations, which indicates the correctness of the conclusion about the change in the zeta potential over time.

The distribution of the current density in the system depends on the resistance of the REM electrode material and the solution. The larger the value of *κ*_m_ at a fixed value of *κ*_s_, the smaller the potential drop at the electrode. In our case, the values of *κ*_m_ and *κ*_s_ are comparable and close to 1.3 S/m, and the potential drop between the boundaries of the electrode is about 0.5 V. Likewise, the current distribution depends on the PCT concentrations due to different gas bubble volume fractions in the bulk of REM (Figure 6).

At the inlet and outlet of the REM, the external interface increases the active surface area. Because the values of *κ*_m_ and *κ*_s_ are comparable, the electrochemical reactions of the hydroxyl radicals and oxygen evolution proceed intensively throughout the entire volume of the REM, and as they approach the boundaries (coordinates 0 and *d*), they significantly increase (Figure 7). At the inlet of the REM, the reaction rates reach their maximums, which causes a sharp decrease in the concentration of PCT during the oxidation reaction with the hydroxyl radicals and an increase in the formation of bubbles due to the high oxygen concentration in the REM (Figure 8).

### 3.7. Influence of Transmembrane Pressure on the Permeate Flux

With a decrease in the transmembrane pressure, the rate of the paracetamol delivery to the active zones of the anode decreases, but the rate the HO^•^ generation is constant. The convective flux of the dissolved oxygen also decreases, as a result of which larger bubbles are formed. This leads to a significant increase in the *ε*_g_ and a decrease in the resulting permeate flux (Figure 9). At the same time, the efficiency of the PCT mineralization increases significantly because of the increase in its residence time in the REM.

The opposite effect is observed with an increase in the transmembrane pressure: a larger solution flow leads to a decrease in the residence time of PCT, as well as a faster removal of dissolved oxygen, which decreases *ε*_g_ and increases the permeate flux.

The presence of bubbles reduces the flow rate of the solution. This phenomenon greatly affects the performance of anodic oxidation systems. In the presence of bubbles, the efficiency of the mineralization increases because the residence time of the organics increases, but the degradation performance decreases because most of the OH^•^ radicals are spent on the formation of molecular oxygen.

## 4. Conclusions

We proposed a one-dimensional time-dependent model of the transport of diluted species and gas bubble evolution in the electrolysis system with the REM. The model is based on Fick’s and Darcy’s laws, and considers the hydrodynamic properties of the system as well as the chemical reactions related to the oxidation of organic compounds by hydroxyl radicals and oxygen evolution reaction. The mass transfer coefficient between the gas and solution phases, and volatility were considered as two crucial parameters.

It was shown that the consideration of the bubble evolution on the surface of the REM pores and electroosmotic flow makes it possible to describe the time variation of the permeate flux with high accuracy in the absence or presence of organic components in the solution. Oxygen bubbles form during the first 15 min of the experiment; after that, their size remains constant until the electric current is turned off. It was found that when an electric current is applied, the zeta potential of the REM pore surface changes with time, which leads to a decrease in the permeate flux due to a decrease in the electroosmotic flow.

In the presence of PCT, the behavior of the system slightly changes: the absolute value of the zeta potential decreases more significantly, and the rate of oxygen generation during the recombination reaction of the HO^•^ decreases. In the presence of PCT of low concentration (17.7 mgC/L), after 1.5 h of the experiment, the permeate flux decrease is comparable to the case of a solution without PCT. This is due to a drop in the electroosmotic flow caused by the decrease in the absolute value of the zeta potential. The reduction in the oxygen generation is insignificant because the PCT concentration is low, which insignificantly reduces the amount of HO^•^ involved in the recombination reaction. At a higher concentration of PCT (141 mgC/L), after 2 h of the experiment, the permeate flux becomes higher than in the case of a solution without organics. This is due to a deceleration in the rate of the bubble formation due to a reduction in oxygen generation from the recombination reaction of HO^•^ because most of them are spent on the degradation of the PCT.

The developed model has predictive ability, allows the identification of the processes responsible for changing the behavior of the system, and can be used to predict the behavior of electrolysis systems with REM.

## Figures and Tables

**Figure 1 ijms-22-05477-f001:**
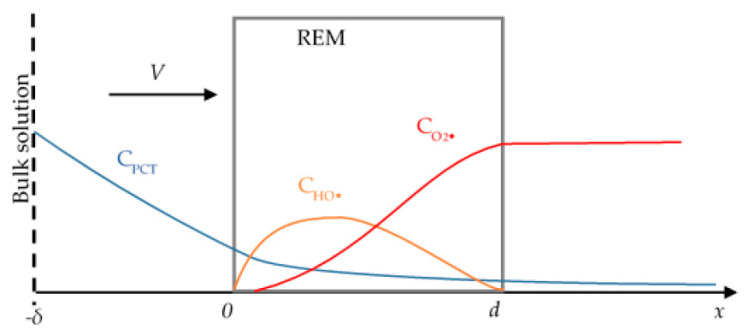
Schematic representation of the system under study. The arrow indicates the direction of the solution flow. Blue line is the spatial distribution of paracetamol concentration (*c*_PCT_), orange line is the spatial distribution of hydroxyl radicals concentration (*c*_HO·_) and the red line is the spatial distribution of molecular oxygen concentration (*c*_O2_).

**Figure 2 ijms-22-05477-f002:**
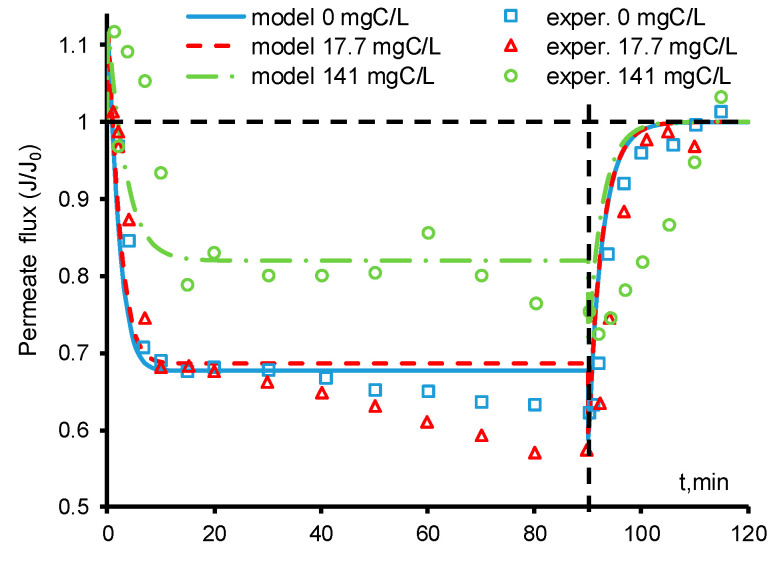
Calculated (lines) and experimental (points) time dependences of the permeate flux. The concentration of the paracetamol is indicated in the figure. The other parameters are presented in Table 1. The experimental data are adapted from [30].

**Figure 3 ijms-22-05477-f003:**
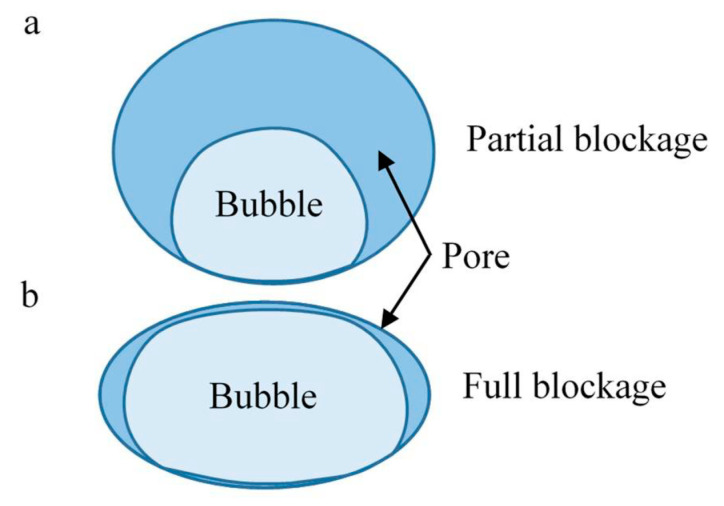
Schematic representation of the placement of a bubble in the volume of a pore with hydrophilic walls. The solution phase is marked in blue and the gas phase is marked in light blue. The partial blockage (**a**) and the full blockage of the pore by bubble (**b**) are presented.

**Figure 4 ijms-22-05477-f004:**
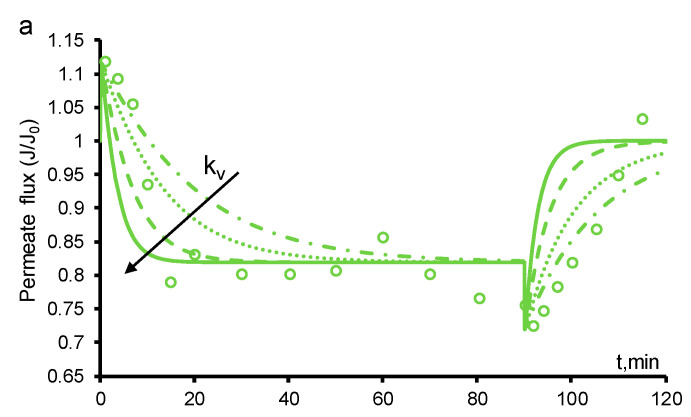
Calculated time dependences of the permeate flux at different parameters *k*_v_ = 3 × 10^−3^ (dashed-dotted), 6 × 10^−3^ (dotted), 10^−3^ (dashed), 1.5 × 10^−3^ (solid) 1/s (**a**) and *k*_g_ = 0.03 (dashed), 0.07 (solid), 0.1 (dotted) m^3^/mol (**b**). The arrows indicate the direction of the increment in the parameters and the circles indicate the experimental data. The paracetamol concentration is 141 mgC/L. The other parameters are presented in Table 1. The experimental data are adapted from [30].

**Figure 5 ijms-22-05477-f005:**
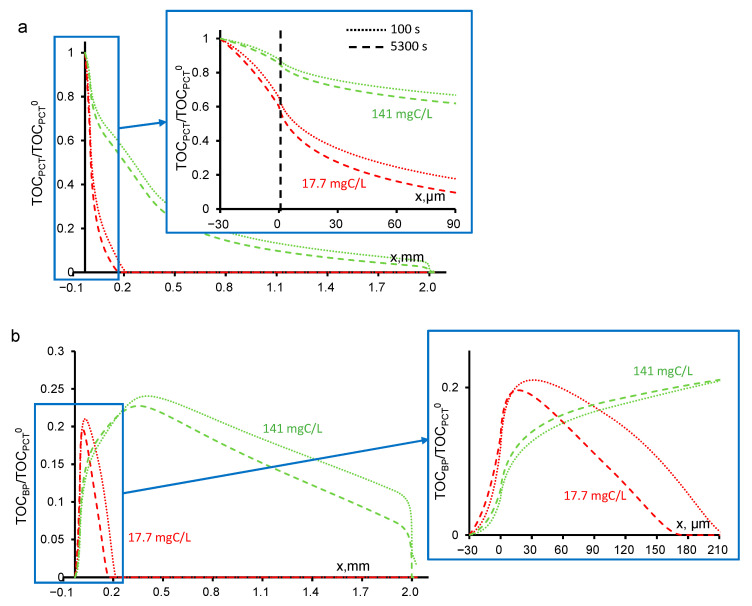
Calculated concentrations of paracetamol (**a**) and byproducts (**b**) in the system under study at different times (100 s, dotted line; 5300 s, dashed lines). The concentration of paracetamol is indicated in the figure. The other parameters are presented in Table 1.

**Figure 6 ijms-22-05477-f006:**
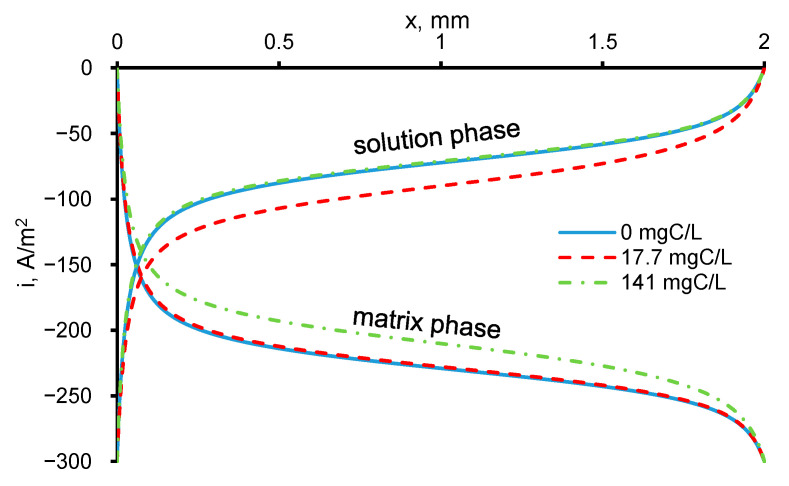
Calculated values of the current density in the solution and matrix phases in the system under study at t = 5300 s. The concentration of the paracetamol is indicated in the figure. The other parameters are presented in Table 1.

**Figure 7 ijms-22-05477-f007:**
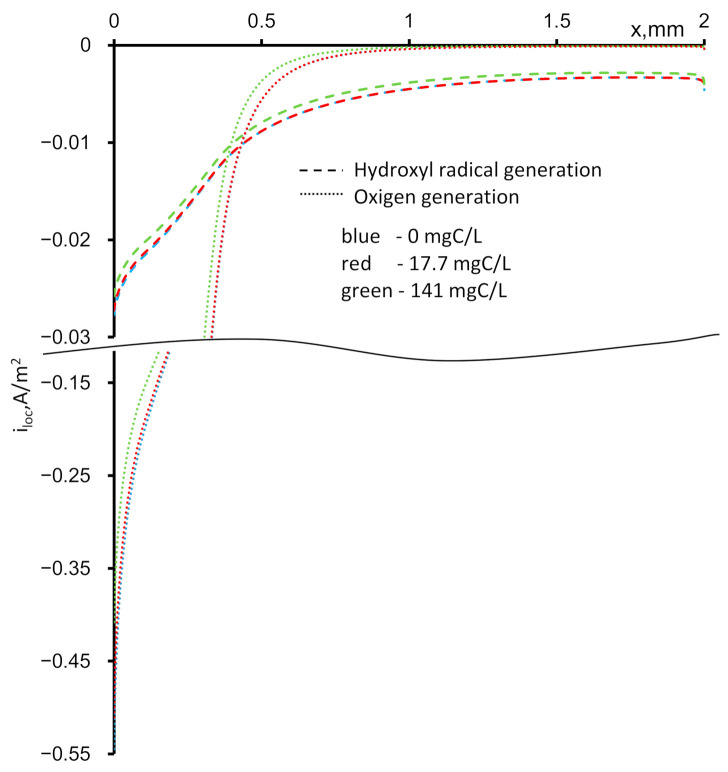
Calculated dependences of the local current densities of the HO^•^ generation (dashed) and oxygen evolution (doted) reaction on the depth of the REM at t = 5300 s. The concentration of the paracetamol is indicated in the figure. The other parameters are presented in Table 1.

**Figure 8 ijms-22-05477-f008:**
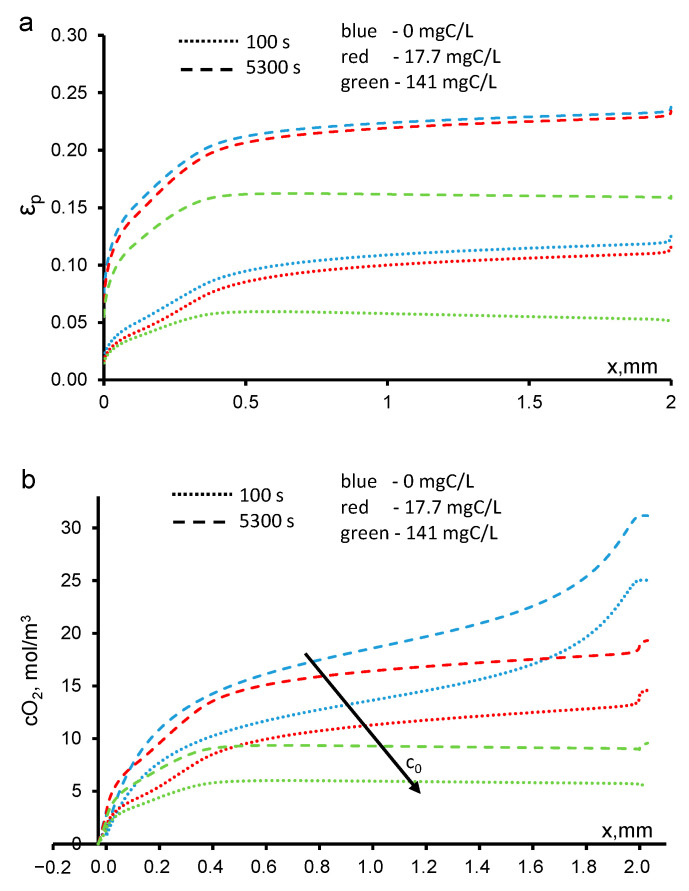
Theoretical dependences of the fraction of the REM volume occupied by bubbles (**a**) and the O_2_ concentration (**b**) on the REM depth. The paracetamol concentration and time are shown in the figure. The arrow indicates the direction of the increment in the paracetamol concentration, c_0_. The other parameters are shown in Table 1.

**Figure 9 ijms-22-05477-f009:**
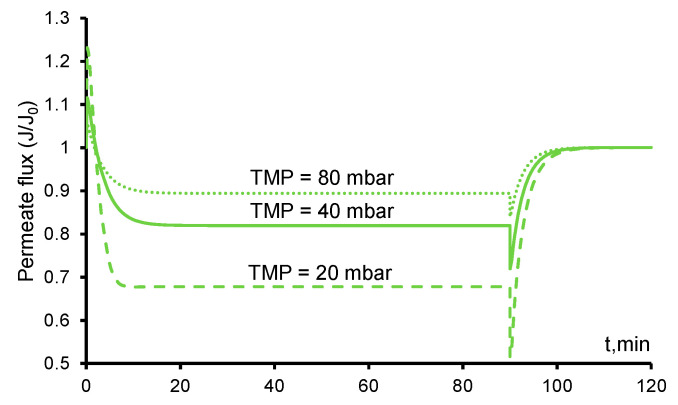
Calculated time dependences of the permeate flux at various transmembrane pressures. The PCT concentration is 141 mgC/L. The transmembrane pressure (TMP) values are shown in the figure near the lines (80 mbar—doted, 40 mbar—solid, 20 mbar—dashed). The other parameters are presented in Table 1.

**Table 1 ijms-22-05477-t001:** Parameters of the system used in the calculations.

*k*_PCT_, mol·s^−1^m^3^		*k_BP_*, mol·s^−1^m^3^		*k*_HO•_, mol·s^−1^m^3^	*D*_BP_, 10^−9^ m^2^/s	*D*_PCT_, 10^−9^ m^2^/s	*D*_HO•_, 10^−9^ m^2^/s	*D*_O2_, 10^−9^ m^2^/s	*μ*, 10^−4^ Pa × s*,*		***ε***
1 × 10^7^ [29,30,31,32]		1.2 × 10^7^ [33]		5.5 × 10^6^ [34]	2.7 × 10^−10^ [35]	0.65 [36]	2.2 [21]	2.2	8.9		80
***κ_s_***, **S/m**	***κ_m_***, **S/m**	***σ***, **m^2^**	***δ***, **μm**	**TMP**, **mbar**	***d***, **mm**	***ε_p_***	***a_v_***, ***10*^8^·1/m**	***n_HO•_***	***n_O2_***	***z_BP_***	***i_tot_***, **A/m^2^**
*1.3* [30]	*1.3* [30]	1.6 [30]	30 [25]	40	2 [30]	0.41 [30]	1.7 [30]	1	4	27	−300
***β***	***k_v_*** **1/s**		***k_g_***, **m^3^/mol**		$E_{{\mathrm{HO}}^{•}}^{0}$, **V**	$E_{O_{2}}^{0}$, **V**	$i_{0}^{{\mathrm{HO}}^{•}}$, **A/m^2^**	$i_{0}^{O_{2}}$, **A/m^2^**		**ζ**, **mV**	
0.5 *	6 × 10^−4^ *		0.07 *		1.7 *	2.05 *	−10^−6^ *	−10^−6^ *		−22 *	

* Fitting parameters.

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
