# Peer review of "Modeling the Formation of Gas Bubbles inside the Pores of Reactive Electrochemical Membranes in the Process of the Anodic Oxidation of Organic Compounds"

_ijms, 2021, doi:10.3390/ijms22115477_

Round 1
Reviewer 1 Report
Work very well written. It presents an interesting topic of research, unpopular in the literature but worthy of recognition.
I believe that the authors only need to emphasize in wnisoakch and in the introduction the aim of the work and its implementation! Additionally, I believe that the experimental part should be described more clearly. The rest is professionally described.
The work deserves to be published
Author Response
Please sea the attachement.

Reviewer 2 Report
The organic molecules treatment is an important field of modern industry development due to the growing environmental problem in the world. Development of efficient removal of organic compounds methods is needed. In this regard, the work has the necessary relevance, because it could help to effectively predict the behavior of REM in flow-through mode. However, before the article publication, there are several comments that could be recommend to pay attention to:
Line 61: “[Bubbles] are formed as a result of water discharge when the electrode potential exceeds the standard electrode potential for oxygen evolution reaction”. Is it correct? The standard electrode potential for oxygen evolution reaction is a thermodynamic quantity equals to 1,23 V. But the real oxygen evolution potential (OEP) for each electrode depends on its nature. And the gas bubbles generation at the anode surface starts when the anode potential exceeds the OEP, not the standard electrode potential. This fact allows all electrodes to be divided into electrodes with high О2 overpotentials and low О2 overpotentials.
Line 118: The rate constant is written as if in superscript. The same for lines 122, 131, 175.
Line 119: Why did you assume kinetics of this reaction as a second-order rate? Can you confirm this assumption by referring to some previous studies?
Line 134-139: I didn’t understand this part. Why in Eq. (10) the overpotential is defined as the difference between the electrode potential and the standard electrode potential? According (Bard. A. J. Electrochemical methods. Fundamentals and Applications) it should be the difference between the electrode potential and the equilibrium potential. Can you, please, provide some additional explanations in this regard?
Just suggestion. Maybe it would be more convenient for the reader to describe the current mode (namely, that the current turns off after 90 minutes of the experiment) earlier? Not directly in the text of discussion, but in “Mathematical model” or in “The experimental data”?
Please, double check the text for mistakes. For example, line 55 contains a mistake in the verb usage. Perhaps, line 156 contains terminological mistake. Flux is already defined as the flow rate per unit area so it is usually used without word “density”.
It would be nice if all citations were in unite style. Consider using [1,2] instead of [1],[2] and [3-5] instead of [3],[4],[5].
